# MCT/LCT Mixed Oil Phase Enhances the Rheological Property and Freeze-Thawing Stability of Emulsion

**DOI:** 10.3390/foods11050712

**Published:** 2022-02-28

**Authors:** Jiahao Liu, Yi Han, Jiashi Chen, Zhigang Zhang, Song Miao, Baodong Zheng, Longtao Zhang

**Affiliations:** 1College of Food Science, Fujian Agriculture and Forestry University, Fuzhou 350002, China; ljiahao9654@163.com (J.L.); hy15514863550@163.com (Y.H.); chenjiashi63@163.com (J.C.); zbdfst@163.com (B.Z.); 2China-Ireland International Cooperation Centre for Food Material Science and Structural Design, Fuzhou 350002, China; song.miao@teagasc.ie; 3State Key Laboratory of Food Safety Technology for Meat Products, Xiamen Yinxiang Group Co. Ltd., Xiamen 361100, China; 18359288677@163.com; 4Teagasc Food Research Centre, Moorepark, Fermoy, P61C996 Co. Cork, Ireland

**Keywords:** emulsion, medium-chain triglyceride, long-chain triglyceride, rheological properties, freeze-thaw stability

## Abstract

The main objective of this study was to investigate the effect of different oil phase compositions (medium-chain triglyceride (MCT) and long-chain triglyceride (LCT), the proportion of MCT is 0%, 5%, 10%, 15% and 20%, respectively) on the rheological properties and freeze-thaw stability of emulsions. The emulsions were characterized by differential scanning calorimetry (DSC), rheometer, stability analyzer, Malvern particle size meter and confocal microscope. Results showed that all emulsions exhibited a gel-like characteristic with a storage modulus higher than the loss modulus. The elastic modulus and complex viscosity of the emulsions increased with the increase of MCT proportions. During the heating from 4 °C to 80 °C, the complex viscosity of all emulsions decreased first and then remained unchanged at a continuous high temperature, indicating that the emulsions had good stability and internal structural integrity during the cooling and high-temperature processes. With the increase of MCT proportions, the freeze-thaw stability of the emulsions increased first and then decreased, and showed the optimum with 10% MCT. That could be referred for the production of a product with better freeze-thaw stability and rheological property in the food and cosmetic industries.

## 1. Introduction

Fat is generally considered associated with obesity and cardiovascular disease, and low-fat foods are of increasing interest [1]. Reducing fat content in foods through various technical means, and designing new food microstructures to mitigate the effects of fat reduction on food texture and flavor have also been researched hotspots in recent years [2]. Emulsions are often used to reduce the fat content of foods and to design smooth-tasting, low-calorie foods [3].

The proportional blending of different fatty acids has specific health effects, which are recognized by the food industry and the nutritional science community [4,5]. Compared with long-chain triglyceride (LCT), fats composed of medium-chain triglyceride (MCT) have lower calories and are more easily absorbed, which have special nutritional or medicinal value [6]. Lipid concomitants had higher cellular antioxidant capacity in MCT than in LCT, and MCT/LCT emulsions were used to extract local anesthetics from serum more effectively than LCT alone to reduce the risk of liver dysfunction [7,8]. Therefore, it is generally accepted that the use of MCT/LCT in combination has potential nutritional health value.

Low-temperature refrigeration is a conventional method to extend the shelf life of food. However, most of the emulsions will undergo physicochemical changes during freezing and thawing, such as fat crystallization, continuous phase freeze concentration and macromolecular conformational changes, which eventually leads to demulsification [9,10]. The size, state and fat composition of the oil phase, as the oil droplets nucleate the emulsion, affect the interfacial structure, stability and rheological behavior of the emulsion, which in turn affects its application scenario and effectiveness [11]. Different chain length fatty acids react differently to temperature changes. The size, quantity and location of crystallization are different during cooling, which has different effects on the freeze-thaw stability of the emulsion [12]. It could be inferred that the freeze-thaw stability of emulsions could be regulated by changing the proportion of MCT and LCT. In addition, the blending of different fatty acids has positive health effects [4,5]. Intake of blended MCT-LCT has potential nutritional health value.

The application of mixed oil phases in the industry is also gradually increasing, and mayonnaise has been a regular food condiment for a long time; while there is a minimal report on the effect of their proportional variation on the physicochemical properties of emulsions. Octenyl succinic anhydride (OSA) modified starch is widely used as an oil-in-water emulsifier and its stable emulsion has good physicochemical properties [13], but few studies have been reported on starch-based emulsions with a mixed oil phase of MCT/LCT.

In this paper, OSA modified starch was used as an emulsifier. Coconut oil and soybean oil were used as models of MCT and LCT, respectively. The effects of different proportions of MCT/LCT on the rheological properties and freeze-thaw stability of emulsion were studied.

## 2. Materials and Methods

### 2.1. Materials

Soybean oil was purchased from Yong Hui Supermarket (Fuzhou, China); commercial OSA starch (HI-CAP 100) was from Ingredion Limited (Shanghai, China); coconut oil was provided by the Wenchang Yefu Company (Wenchang, China).

### 2.2. Preparation of Stock Modified Starch Stabilized Emulsions

OSA starch was dissolved in deionized water and stirred gently (500 rpm) for 2 h using a magnetic stirrer. Coconut oil was added to the soybean oil so that the proportion of coconut oil in the mixed oil phase was 0, 5, 10, 15 and 20 wt%, respectively, and then heated to 50 °C and held for 1 h until the two were mixed well.

Mixed fats in different proportions were subsequently mixed with the OSA starch dispersion at ambient temperature. The ratio of the oil phase to the aqueous phase in the emulsion was 40:60 (*w/w*), with a final OSA starch concentration of 4 wt%. The mixed liquid is treated with a high-speed shearing disperser (IKA, T18DS25, Staufen, Germany) for 5 min (13,000 rpm) to obtain a coarse emulsion, in reference to the method of Kobayashi et al. [14]. The obtained coarse emulsions were sonicated separately in an ultrasonic cytometer (CSIENTZ-950E, Ningbo, China) under the following conditions: ultrasonic power of 300 W, working for 4 s, stopping for 2 s, and a total ultrasonic time of 10 min. The obtained fine emulsion was stored at 4 °C.

### 2.3. Differential Scanning Calorimetric Analysis of Mixed Oil

A differential scanning calorimeter (DSC, TAQ2000, Selb, Germany), equipped with a thermal analysis data station, was used. Nitrogen (99.999% purity) was the purge gas and flowed at ~20 mL/min. Samples of 6–12 mg were transferred into aluminum pans sealed. All samples were subjected to the following temperature program: 50 °C held for 5 min, cooled at 5 °C/min to −80 °C and held for 5 min. The same sample was then heated from −80 °C to 80 °C at the same rate [15]. The crystallization characteristics of each sample in a DSC scan can be indicated at various temperatures.

### 2.4. Small Deformation Rheology

We used the Anton Paar rotary rheometer (MCR301, Graz, Austria), equipped with a flat plate geometry system (Model CP50), with a plate spacing of 0.5 mm. An appropriate amount of emulsion sample was placed in the center of the rheometer plate and excess samples around the plate were removed to avoid the effect of edge-effect. In order to avoid sample water loss during the test, the conical plate was sealed using silicone oil.

First, the emulsion was scanned at the rate of 0.01–100 s^−1^, and the viscosity and shear stress of the sample were recorded. Secondly, the elastic modulus (G′) and viscous modulus (G″) were measured while conducting a strain sweep between 0.01% and 100% strain, at 1 Hz and 25 °C, to determine the linear viscoelastic region. Then, in the linear viscoelastic region, 0.5% strain was selected and a frequency sweep of 1–100 Hz was performed to record the elastic modulus (G′) and viscoelastic modulus (G″) of the emulsion. The last experimental content was a temperature scan. In the linear viscoelastic region, 0.5% strain and 1 Hz were selected, the scan temperature range was 4~80 °C, the heating rate was 4 °C·min^−1^, and the compound viscosity of the emulsion during the heating and holding process was recorded.

### 2.5. Freeze-Thawing Treatment

Fifty-milliliter emulsions were frozen in an ultra-low temperature refrigerator (DW-HL3985S, Hefei, China) at −80 °C for 24 h and then thawed in a water bath at 50 °C for 2 h until they were completely thawed.

### 2.6. Stability Analysis

Its stability was analyzed using a stability analyzer (LUMiSizer, Berlin, Germany). We transferred 420 μL of sample to a 2 mm optical diameter PC cuvette. The experimental parameters were set as follows: NIR wavelength 880 mm, centrifugation rate 1500 rpm, centrifugation temperature 25 °C, contour line 300, scan rate 30 s^−1^, and a total centrifugation time of 2.5 h [16].

### 2.7. Particle Size Analysis

The average particle size and particle size distribution of emulsion droplets before and after freeze-thawing were determined by a laser particle sizer (Mastersizer3000, Nottingham, UK). Optical test mode was set to background test time 15 s, relative refractive index (RI) of emulsion droplets 1.095 (i.e., the ratio of the RI of oil (1.46) to that of the aqueous phase (1.33)). We stirred the sample thoroughly before testing to ensure uniformity of the sample. The stirring speed was 2000 rpm/min for dispersion, and the shading degree was set to 5–18%. We added the sample until the shading degree was about 10%, and took an average of 3 times for each measurement.

### 2.8. Confocal Laser Scanning Microscopy

All emulsions samples were imaged via confocal laser scanning microscopy (CLSM). A Zeiss confocal microscope (LSM 880, Jena, Germany) with a 40× magnification lens was used. About 10 µL of Nile Red (1 mg mL^−1^ in dimethyl sulfoxide, 1:100 *v/v*) was used to stain oil (argon laser with an excitation line at 488 nm), 10 µL of Nile Blue (0.1 mg mL^−1^ in Milli-Q water, 1:100 *v/v*) was used to stain OSA starch (argon laser with an excitation line at 639 nm). Approximately 0.5 mL of sample was mixed completely with 20 μL aliquots of Nile Blue and Nile Red solutions. We took about 20 µL of the stained sample on the glass slide to ensure that there was no gap between the stained sample and the cover glass to prevent air bubbles.

### 2.9. Statistical Analysis

Data were obtained in triplicate and mean and standard deviation was calculated. Charting was with Origin Pro 8.6.

## 3. Results

### 3.1. Thermal Analysis by DSC

Crystallization is always used to characterize the thermal behavior of oil samples. The thermal profiles of five mixed oil samples were examined by differential scanning calorimetry (DSC) and the crystallization profiles and the degree of blending of these oil samples were reported (Figure 1).

The low-temperature region defines the crystalline part of the oleic acid and the high-temperature region defines the crystalline part of the stearic acid, and the crystallization curves for all samples were only for the lower temperature region. Compared to the 0% sample, the 5% and 10% samples show two overlapping exothermic peaks, while the 15% and 20% samples disappear and show three small exothermic peaks. The 0%, 5% and 15% samples showed two distinct exothermic peaks, and the lowest exothermic peak (crystallization endpoint) was sharper than the shoulder peak; the three exothermic peaks (lowest peak and two shoulder peaks) of the 15% and 20% samples were all relatively smooth. The smoother and lower the lowest exothermic peak with increasing MCT addition may be due to the change in the type and content of oleic acid [17]. It may also be due to the fact that the mixed oil phase as a simple composite (where soybean oil and coconut oil are treated as one material) is not well mixed, is strong, and only one of the substances may be analyzed during the experiment so that multiple unsharp peaks appear when the addition amount is increased [18].

All crystallization points are read at the minimum of exothermal peaks. The crystallization point of the control group was −24.01 °C. The crystallization points of 5%, 10%, 15% and 20% were −22.30 °C, −19.08 °C, −9.4 °C and −8.78 °C, respectively, and the crystallization temperature increased abruptly when the addition amount reached 15%. This is because of the high content of saturated fatty acids in MCT, which is easy to crystallize [19]. When the proportion of MCT reached about 15%, it caused a significant change in the crystallization point of the blend.

### 3.2. Rheological Properties of Mixed Oil Phase Emulsions

The 0% sample was set as a control (same below). Figure 2 shows the relationship between the apparent viscosity of emulsions and shear stress with shear rate. As the shear rate became larger, the apparent viscosity of all emulsions showed a decreasing trend, which was typical of pseudoplastic fluids [20].

The apparent viscosity of the emulsions increased sequentially with the increase of the MCT addition, the 20% sample had the largest apparent viscosity, which was due to the increase in saturated fatty acid content in the oil phase of the emulsion as the proportion of MCT increased, so the emulsion oil droplets partially crystallized and made the viscosity increase. Further, the crystallization of the emulsion oil droplets caused more of the emulsifier that was originally wrapped around its exterior to dissolve in the aqueous phase, which led to an increase in viscosity [21]. As the shear rate increases, the shear stress gradually increases, which is the result of the increase in the viscosity of the emulsion. When below the critical applied stress (yield stress), they behave like elastic solids. Above this stress, they behave as viscous liquids and exhibit shear-thinning [22,23].

Figure 3 shows the changes in the elastic modulus (G′) and viscous modulus (G″) of the emulsion sample under strain scanning (0.01~100%), which determines the conditions for frequency scanning.

In the linear range, G′ was greater than G″ for all emulsion samples, indicating that the elastic part of the emulsion was greater than the viscous part within the emulsion, exhibiting gel-like properties. As the MCT addition increased, the emulsion G′ increased and the linear viscoelastic interval narrowed, with the linear interval for the sample of 15% and 20% being smaller than that of the sample of 5% and 10%. After the strain is greater than 1%, G′ starts to decrease sharply for both 15% and 20% samples, while it decreases slowly for 5% and 10% samples only after them. This indicates that when larger stresses are applied, the shape of the droplet starts to undergo irreversible deformation and the internal structure of the emulsion starts to be destroyed, the 15% and 20% samples being more susceptible to the destruction.

The linear viscoelastic interval of all emulsions was determined from the strain scan, and a frequency scan (0.1~10 Hz) was performed by selecting a strain condition of 0.5%, and the results are shown in Figure 4.

For all samples, G′ was greater than G″ and no crossover was observed. In the frequency scan range, the viscoelastic gap of the 0% emulsions remained relatively stable and showed a weak frequency dependence in the mixed oil phase. The 5% and 10% samples showed a similar trend. For the 15% and 20% samples, with the increasing MCT proportion in the oil phase, its viscoelastic gap increased with increasing frequency and the emulsions were more elastic. This may be due to the important role of MCT on the elastic modulus of the emulsion and may further increase the strength of the network structure in the emulsion gel [24].

The thermo-rheological behavior of the emulsion was shown in Figure 5. The complex viscosity is obtained from the ratio of G′ to G″. The composite viscosity of the emulsion samples increased with the addition of MCT during the warming scan. However, the respective composite viscosities of all samples showed a decreasing trend with the increase in temperature.

It could be inferred that fat crystallization in oil-in-water emulsions affects their rheology. With the addition of MCT, the crystallization point of the mixed oil phase increased. The resulting presence of crystals in the emulsion oil phase at room temperature leads to a gradual increase in G′ (G″ remains essentially unchanged), so the composite viscosity increases. In addition, unlike protein-based emulsions that gel significantly after heating, starch modified by OSA is much more hydrophobic and shows strong surface activity, and the emulsions maintain high stability at both low and high temperatures [25]. As the temperature increases, it gradually approaches the melting point of MCT (around 24 °C) until the high temperature is maintained, the crystalline fat is converted and G′ decreases, so the complex viscosity tends to decrease.

Overall, it seems that the composite viscosity of the samples all decreased slowly in the range of 4–80 °C, indicating that the internal structure of the emulsions remained relatively stable during the process. The composite viscosity of the samples remained unchanged within 4 min of constant temperature at 80 °C, indicating that all the emulsions remained stable at a certain time under the continuous state of high-temperature.

### 3.3. Freeze-Thawing Stability of Mixed Oil Phase Emulsions

The macroscopic changes before and after freeze-thawing are shown in Figure 6. There was no significant difference between the emulsion samples before freezing. After freeze-thawing, the emulsion samples were stratified into oil, emulsified and aqueous phase layers. This is due to the crystallization of oil and water in the emulsion droplets during the freezing treatment, resulting in a state transition in the internal structure. When the formation of crystals is too large, the emulsion droplet interface film is punctured, and the oil droplets are aggregated and flocculated after thawing. Finally, due to the relative density difference, the oil phase floats up, the water phase precipitates, and the middle is the emulsion layer [12]. With the increase of MCT, the height of the water phase layer decreases until it disappears (10%), but the oil layer starts to appear (15%). It seems that the emulsion with 10% MCT shows the optimum freeze-thawing stability.

The stability analyzer reflects the variation of the particle delamination rate with position during centrifugation of the emulsion before and after freeze-thawing transmittance. The samples reached the final delamination position within a certain time, and the greater the range of variation in transmittance, the less stable they were [26]. The red line (the first spectral line recorded at the bottom) and the green line (the last spectral line recorded at the top) of all emulsion samples before freezing treatment largely overlapped when they reached the delamination position, indicating that the samples were uniformly stable (Figure 7). After the freeze-thaw treatment, the emulsions were divided into three layers (oil layer, emulsified layer and water layer, as shown in Figure 6). The increase of MCT addition narrowed the transmission range of the aqueous layer, indicating a decrease in the height of the aqueous layer precipitation. Furthermore, 15% and 20% of the samples showed a large number of fast-moving particles at the beginning, i.e., an oil layer, and an increase in the overall transmission range, which made the stability worse. Moreso, 10% of the samples had the best stability after freeze-thawing with a significantly lower aqueous layer height and no oil layer precipitation. This may be due to the fact that the addition of MCT regulates the crystallization behavior of the mixed oil phase, which further affects the freeze-thaw stability of the emulsion. When the MCT addition is around 10%, the reduction of emulsion freezing to crystal size improves the stability; while when the proportion reaches 15%, the crystallization point of the mixed oil phase changes drastically, leading to destabilization of the emulsion after thawing and reducing the stability. Different proportions of MCT replacing LCT in the mixed oil phase may have different effects on the emulsion stabilization stability and sensory properties [27]. In addition, the chain length and content of fatty acids in the oil base also influenced the storage stability of the emulsions [28].

The addition of MCT increases the storage modulus of the samples, and a higher storage modulus means higher stress is required to make the emulsion flow [29]. Therefore, structural changes in all emulsions during frozen storage and after thawing are readily expected. In addition, network rearrangements and even flocculation are expected to occur due to the strong droplet–droplet interactions in thick emulsions. It is believed that emulsions containing 5% and 10% undergo structural rearrangement after thawing, with re-network formation between droplets and improved stability. MCT replacements of 15% and 20% were the most unstable emulsions, and according to Tadros [30], the increase in storage modulus with time was attributed to the formation of strong droplet flocculation. The DSC thermal crystallization curve shows that the crystallization temperature of the oil phase increases with the addition of MCT. The oil phase with the highest crystallization temperature was the first to start forming ice nuclei, and as the temperature continued to decrease, the larger the ice crystals eventually formed [31]. Therefore, the 15% and 20% samples with high crystallization temperature start to form ice nuclei first, and over time, the flocculation is enhanced, and eventually, the formation of oversized ice crystals punctures the interfacial film, breaking the emulsion droplets, and the oil droplets converge after thawing. In the same way, the nucleation time of 5% and 10% samples was late, which eventually formed smaller crystals and showed higher freeze-thaw stability [32].

### 3.4. Characteristics of Emulsion Particles

The composition and physicochemical properties of the oil in the dispersed phase affect the size of the droplets produced in the homogenization process. Before freezing, the particle size of the emulsions with different proportions of mixed oil phases was similar, as shown in Figure 8a. The overall particle size distribution curve shows a “bimodal” pattern, with the range of the bimodal peaks being close to each other and the particle sizes being mostly in the range of 1–10 μm. The above results show that the change in the proportion of MCT did not lead to significant destabilization or agglomeration.

The particle size distributions after freeze-thawing are shown in Figure 8b. The distributions of the samples of 0%, 5% and 10% show “double peaks” between 1–10 μm, while the distributions of the samples of 15% and 20% show “single peaks” between 60–110 μm. With the increase of MCT, the particle size of the 15% and 20% emulsion samples increased significantly, about 10 times more than before freeze-thawing. It could be inferred that the increasing MCT proportion in the oil phase led to larger fat crystal forming during freezing [33]. Moreover, the emulsion particle was damaged, leading to coalescence or flocculation, resulting in larger emulsion particles forming (Figure 8b) and decreasing stability until the emulsion delaminates. As a result, the particle size change and stability analysis of the emulsions before and after freeze-thaw showed consistency.

In order to analyze the droplet status in emulsion, the samples before and after freeze-thawing were quickly captured using microscopy for droplet images after high-speed shearing (Figure 9). As shown in Figure 9, the mixed oil phase emulsions are better stabilized in the order of 10%, 5%, 15 and 20%, which is consistent with the average particle size profile (Figure 8). After freezing and thawing, the emulsion stability first increases and then decreases. The addition of a certain amount of MCT increased the freeze-thaw stability of the emulsions. However, as the MCT proportion continued to increase, when the frozen fat crystals were too large and the interfacial film was damaged, that instead destabilized the emulsion after thawing. After freeze-thawing, 15% and 20% emulsion droplets underwent significant aggregation and particle size increase; 5% and 10% of the emulsions showed no significant change in particle size, and 10% of the samples showed insignificant delamination and improved freeze-thaw stability.

## 4. Conclusions

In summary, an oil-in-water emulsion with MCT/LCT mixed oil phase was prepared, which showed high freeze-thawing stability. The elastic modulus and complex viscosity of the emulsion increased with the increase of MCT proportions. When the proportion of MCT was 10%, the DSC graph of the oil phase demonstrate low crystallization points. Furthermore, after freeze-thawing, 10% MCT emulsion does not exhibit stratification and indicated a stable droplet size and particle size distribution, showing the best freeze-thaw stability. This could be referred for producing a product with better stability and rheological properties in the food industry and cosmetics industry.

## Figures and Tables

**Figure 1 foods-11-00712-f001:**
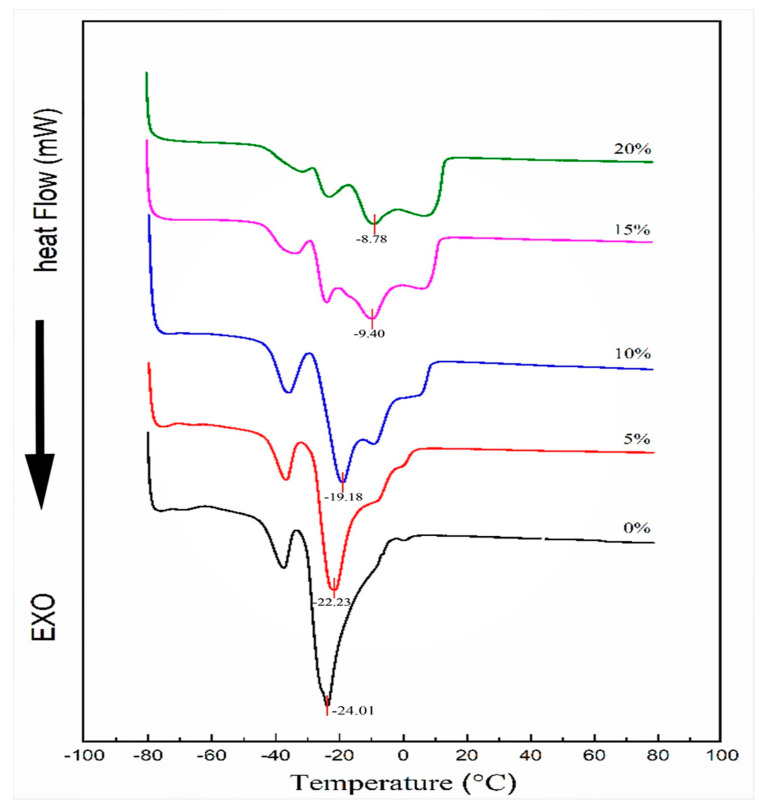
Differential scanning calorimetry crystallization curve of the mixed oil phase.

**Figure 2 foods-11-00712-f002:**
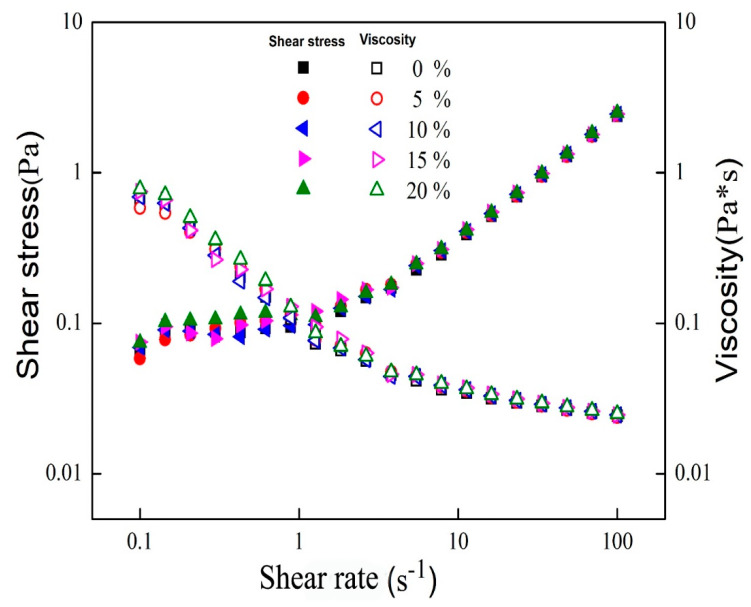
Effect of different proportions of mixed oil on emulsion viscosity and stress.

**Figure 3 foods-11-00712-f003:**
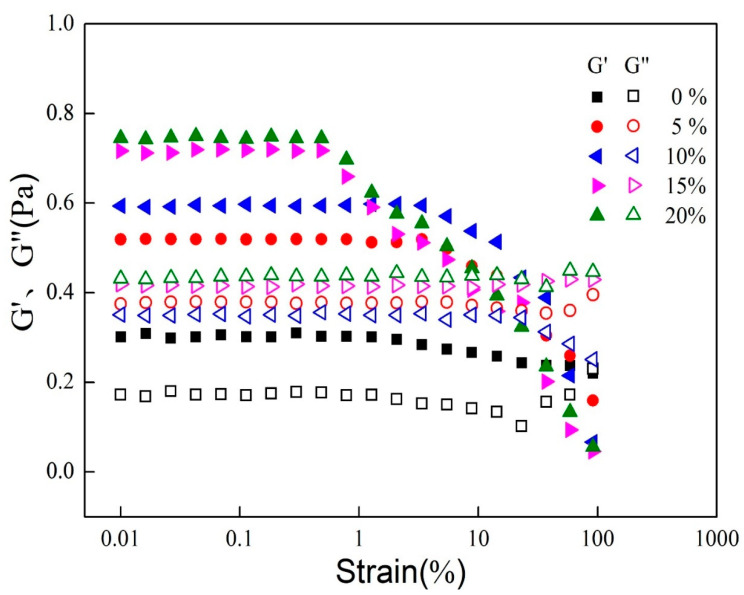
Elastic modulus (G′) and viscous modulus (G″) of emulsion under 0.01–100% strain conditions.

**Figure 4 foods-11-00712-f004:**
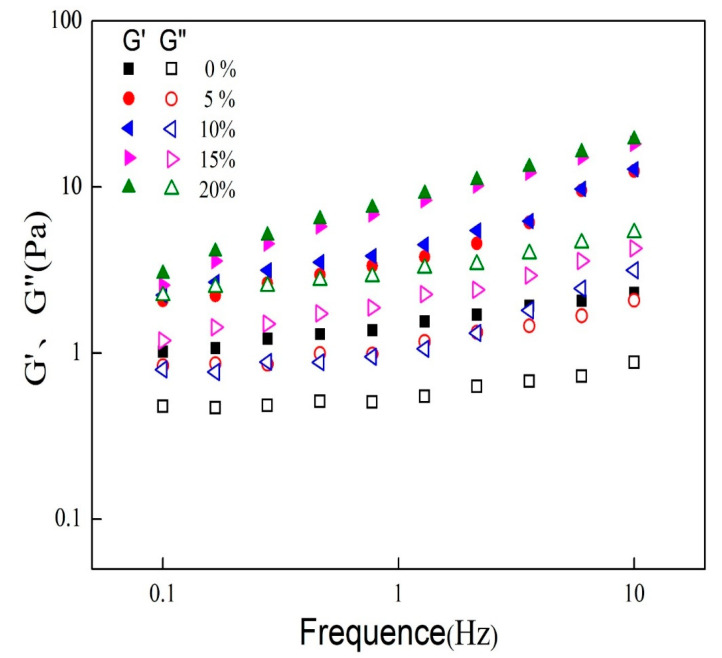
Elastic modulus (G′) and viscous modulus (G″) of emulsions at 0.1–10 Hz.

**Figure 5 foods-11-00712-f005:**
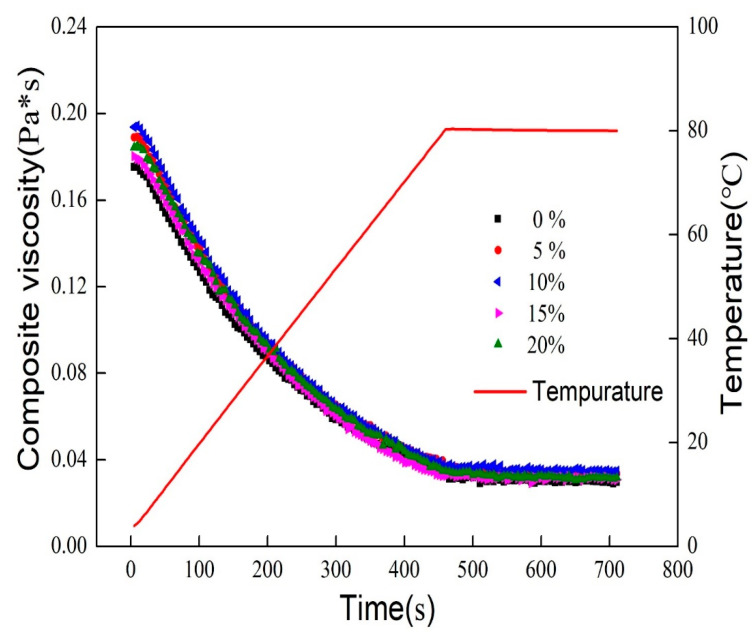
Viscosity-temperature curve of emulsions (4–80 °C).

**Figure 6 foods-11-00712-f006:**
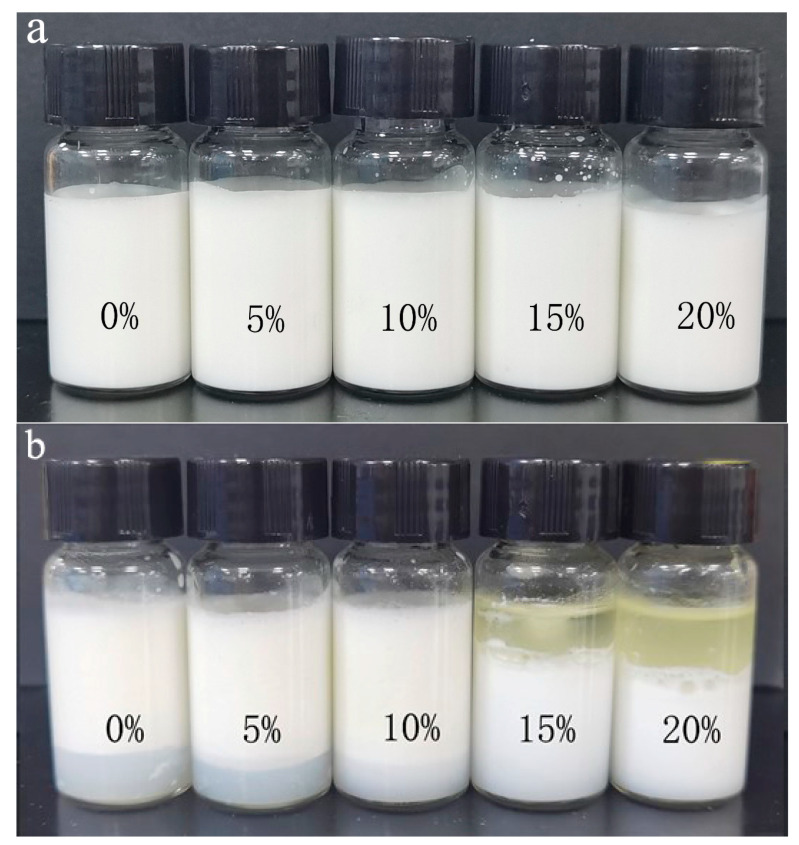
Macrograph of emulsion before freeze-thaw (**a**) and after freeze-thaw (**b**).

**Figure 7 foods-11-00712-f007:**
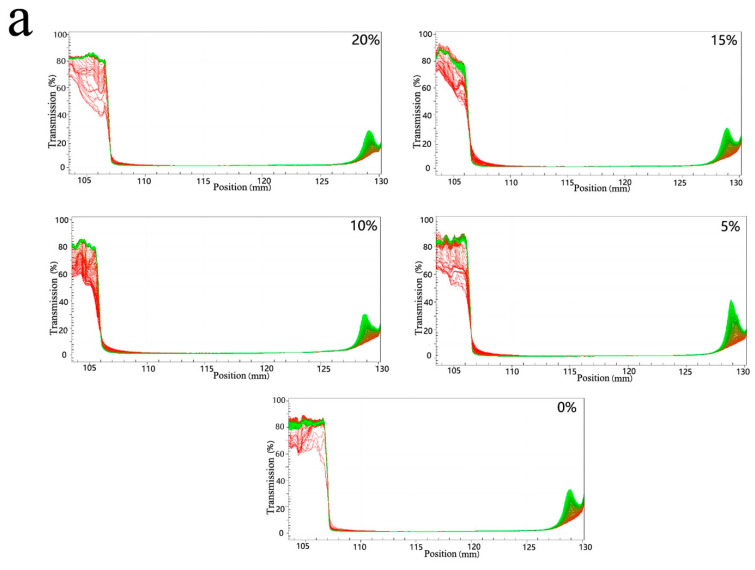
Stability profiles of emulsions before freeze-thaw (**a**) and after freeze-thaw (**b**).

**Figure 8 foods-11-00712-f008:**
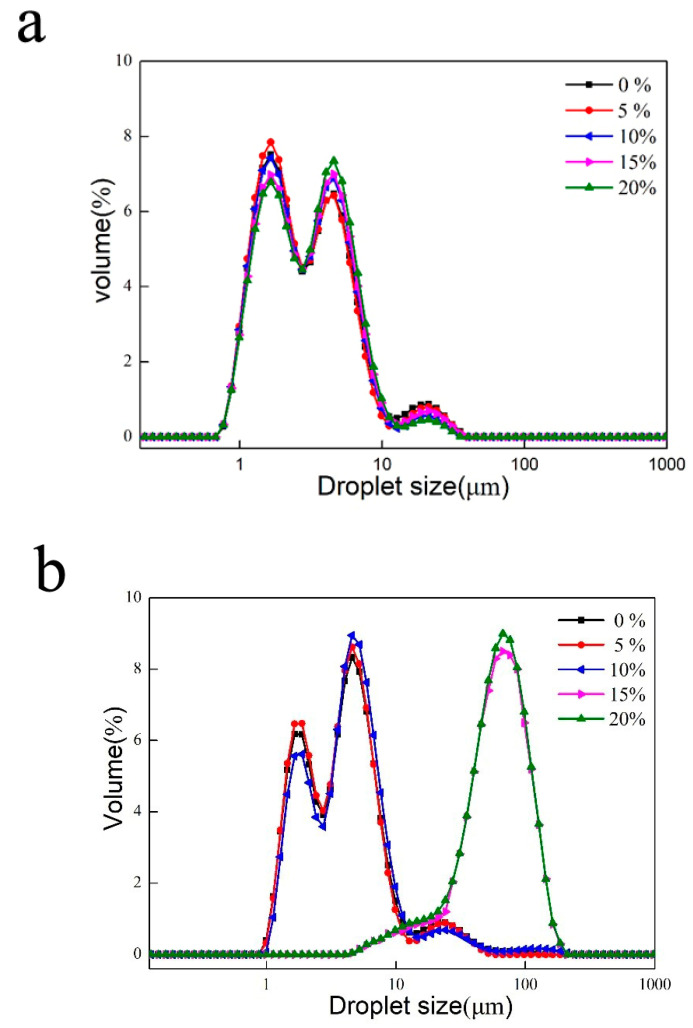
Distribution of emulsion particle size before freeze-thaw (**a**) and after freeze-thaw (**b**).

**Figure 9 foods-11-00712-f009:**
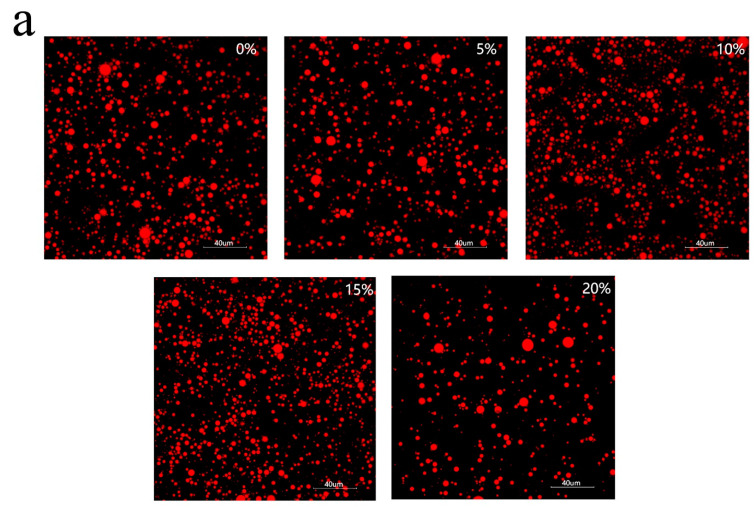
Microstructure of emulsion before freeze-thaw (**a**) and after freeze-thaw (**b**).

## Data Availability

The data presented in this study are available on request from the corresponding author.

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
