# Peer review of "MCT/LCT Mixed Oil Phase Enhances the Rheological Property and Freeze-Thawing Stability of Emulsion"

_foods, 2022, doi:10.3390/foods11050712_

Round 1
Reviewer 1 Report
|
Comments 1 |
Overall the paper is of really good quality with an impressive domain of work, and is worthy to get published. Just elaborate the significance of use of emulsions in daily life? Can these be a suitable replacement of dietary fats in light of health perspective? |
|
Comments 2 |
Line 68: It is stated that blending of fatty acids have “specific health effects”, It should be elaborated that these effects are positive or negative for regular consumption? |
|
Comments 3 |
Line 82 & 83: Avoid unnecessary spacing between brackets and semicolons. |
|
Comments 4 |
Line 146: Please explain abbreviations that are not common to readers like “HeNe” |
|
Comments 5 |
Line 267: Only 10% samples; are not these very little? How can it be recommended for industrial use? |
|
Comments 6 |
Line 309: Its is claimed that: “the particle size change and stability analysis of the emulsions before and after freeze-thaw showed consistency” How can you justify the changing particle size will show stability in results? |
|
Comments 7 |
Line 319: In conclusion please add some practical applications of your work done for oil industry. |
Author Response
Shown in the submitted file

Reviewer 2 Report
The article is interesting and has potential, but also has weaknesses. The article requires professional correction of the writing style. The authors should use the MDPI article template before sending the manuscript to Foods.
Abstract
The abstract needs improvement. It is not clearly presented. Writing style is incorrect! Without reading the entire text, you don't know what's inside.
Introduction
The introduction requires improvement, presenting the current state of knowledge about the previous research.
L70-71 and L74-75. Add information and literature about previous research. What was found?
L72-74. More studies of stability of emulsions with an addition of OSA starch should be add.
L76-79. What was the purpose of the research?
Material and methods
L92. How was the OSA starch concentration selected?
L95-97. What device generated ultrasound? Was it an ultrasonic homogenizer? Was the temperature of the emulsion tested during homogenization? The material becomes warm when ultrasound is applied. This may have a negative effect on the physicochemical properties of the emulsion.
L96. ‘300 w’ W is written in capital letters.
Results and Discussion
L199. “The last is a temperature scan” What do you mean?
L124-125. What device was used to freeze the emulsion? What was the mass of the frozen sample?
L127-130. Please add more details. What was the sample volume? The emulsion samples were transferred into glass bottles and measured stability? And then stored in a freezer and thawed? What was the point of reference in the case of freeze-thawing emulsion (Fig. 7b)?
L132. Average particle size of oil droplets is not given in Results section.
L137. It should be “2000 rpm/min”.
L138-139. Were these 3 independent measurements? Or 1 sample measured 3 times?
L143-146. In what proportion was the dye added to the emulsion?
L150. The presented data (figures) do not include the standard deviation. Were the slopes of the curves statistically compared?
L174. Blank group? What do you mean?
L178-179. Too large? What do you mean? It should be clearly stated.
L181. ‘Taking the 0% sample as a blank control (the same as below).’ Style!
L182-184. Please add literature.
L213. Blank emulsions? What do you mean?
L258-261. What about the difference in transmissions in position 105 mm within a given emulsion (Fig. 7)? Is emulsion stable before freezing?
L262-263. Figure 8? Please check.
L268-270. Explain the impact by developing this sentence. Discussion is needed.
L270-272. Explain the impact by developing this sentence. Discussion is needed.
L293-318. What about the influence of particle droplet size on emulsion stability? Discussion is needed.
L316-318. Discussion is needed.
Conclusion
Conclusions need improvement. They are too laconic.
Figure 2. In the title of the OX axis, add a “shear rate”
Author Response
Shown in the submitted file

Round 2
Reviewer 2 Report
Statistical analysis. The presented data does not contain standard deviations. Are the curves in Fig. 1. and Fig. 7 for individual emulsion represent the mean value of 3 replicates? Are there statistically significant differences in the course of individual curves?
Conclusions need improvement, "the freeze-thaw stability first strengthened and then weakened" what does it mean, what does it bring to research? How different proportions of MCT/LCT effect on viscosity, stress, and particle size? It is also worth adding conclusions from the DSC.
Other comments:
Line 21 and elsewhere. Standardize the notation of the temperature unit, there should be a gap between the numerical value and the measurement unit.
Line 67. Remove the bracket.
Line 86. Delete the repetition. "... under the following conditions: under the conditions: ..."
Lines 142-144. The presented data does not contain standard deviations. Are the curves in Fig. 1. and Fig. 7 for individual emulsion represent the mean value of 3 replicates?
Line 225. Correct the title of Fig.5. Write that it is about emulsions.
Line 261-263. As evidenced by the increased values of transmittance in time (2.5 h) (Fig. 7)? From what point did the emulsions reach their "delamination position"?
Line 344-345. The OSA was not mentioned in abstract and the title does not contain that expression; it is proposed to remove “OSA emulsion” and save it as an “emulsion”.
